# Minor Variety of Campania Olive Germplasm (“Racioppella”): Effects of Kaolin on Production and Bioactive Components of Drupes and Oil

**DOI:** 10.3390/plants12061259

**Published:** 2023-03-10

**Authors:** Aurora Cirillo, Giulia Graziani, Lucia De Luca, Marco Cepparulo, Alberto Ritieni, Raffaele Romano, Claudio Di Vaio

**Affiliations:** 1Department of Agricultural Sciences, University of Naples Federico II, Via Università 100, 80055 Portici, Italy; 2Department of Pharmacy, University of Naples Federico II, Via Domenico Montesano 49, 80131 Naples, Italy

**Keywords:** *Olea europaea* L., anti-transpirant, total polyphenols, fatty acids composition

## Abstract

The effects of climate change have a great impact on the Mediterranean regions which are experiencing an increase in drought periods with extreme temperatures. Among the various solutions reported to reduce the damage caused by extreme environmental conditions on olive plants, the application of anti-transpirant products is widespread. In an increasingly current scenario of climate change, this study was designed to evaluate the effect of kaolin on the quantitative and qualitative parameters of drupes and oil in a little-known olive cultivar known as “Racioppella”, belonging to the autochthonous germplasm of Campania (Southern Italy). To this purpose, the determination of maturation index, olive yield/plant, and bioactive components analysis (anthocyanins, carotenoids, total polyphenols, antioxidant activity, and fatty acids) were carried out. Kaolin applications showed no statistically significant differences in terms of production/plant while a significant increase in the drupe oil content was observed. Kaolin treatments resulted in increased anthocyanins (+24%) and total polyphenols (+60%) content and at the same time a significant increase in the antioxidant activity (+41%) of drupes was recorded. As far as oil is concerned, the results showed an increase in monounsaturated fatty acids, oleic and linoleic acids, and total polyphenols (+11%). On the basis of the results obtained, we can conclude that kaolin treatment can be considered as a sustainable solution to improve qualitative parameters in olive drupes and oil.

## 1. Introduction

Olive tree cultivation is one of the greatest Italian treasures [1,2], it is an arboreal fruit crop typical of the Mediterranean basin where it covers over eight million hectares. It is estimated that the Italian germplasm olive tree includes about 600 varieties, most of which are indigenous and propagated vegetatively at farm level since ancient times [3], and new local genotypes are continuously described. Italy holds the world record for the number of cultivated varieties, representing 25% of the olive germplasm known in the world [4]. Varieties represent the most important strategic element to boost the commercial value of Italian extra virgin olive oil [5]. Olive cultivars are believed to be varieties of unknown origin presumably selected by growers over the centuries from wild genotypes, perhaps those producing the larger fruits. Hence, the high genotypic diversity of olive varieties could be explained by human selection in response to local environmental and agronomic conditions [6,7,8]. It is also likely that crosses between wild local olive genotypes and introduced cultivars have occurred in many areas, thereby leading to new cultivars. In Italy, each region has its own local cultivars, and many seedling trees grow spontaneously [9]. The larger olive-producing Italian regions are Apulia, Calabria, Sicily, and Campania. The use of *Olea europaea* L., as both table and oil cultivars, is well-documented in these regions, with many archaeological and written relics dating back to ancient times, attested also by the presence of monumental trees.

Thanks to its varied pedo-climatic conditions, Campania has a very rich gene pool of olive cultivars [10]; its oil production is based mainly on local varieties, found on approx. 80,000 small farms distributed throughout the region. In many areas, olive cultivation remains a key element of the rural economy and soil conservation strategies. Five PDO (Protected Designation of Origin) labels such as “Cilento”, “Colline Salernitane”, “Penisola Sorrentina”, Irpinia-Colline dell’Ufita”, and “Terre Aurunche” have been granted for the extra-virgin olive oil produced in Campania and a PGI (Protected Geographical Indication) label called “Olio Campania” [10]. However, little is known about the minor olive cultivars which play equally important roles both from a nutraceutical and agronomic point of view as already documented in our previous study on minor varieties such as “Oliva Bianca” [11]. Despite the growing interest in high-quality olive oils, there are very few studies related to the olive drupes and oils of Campania minor cultivars as most of the works in literature are studies on major Italian cultivars. The study and recovery of minor cultivars existing in specific cultivation areas is very attractive and could represent a valid contribution to improve the range of products offered to consumers [12]. Olive diversity can be assessed by considering different characters which include morphological and agronomic traits [13], allozyme variations [14], DNA polymorphisms [13,15], and their combination [10]. Most of the traditional cultivars are exclusively used for oil production; only a few cultivars, such as “Ortice”, “Ortolana”, and “Caiazzana” are suitable for use both as table olives and oil production. Several local cultivars are thought to be resistant to cold (e.g., “Rotondella”) or drought (e.g., “Carpellese” and “Pisciottana”) conditions, and a few cultivars are putatively considered to be resistant to black mold, peacock spot, and knot diseases (e.g., “Pisciottana”) [9]. The cultivars respond differently to high-temperature and drought conditions showing differences in adaptation, vitality and germinability of pollen, dry matter distribution, production, and gas exchange responses to water shortage [16,17,18,19]. In this context, climate change may become particularly challenging for olive growers, for this reason it is necessary to adapt different measures to mitigate their effect on crops. To effectively cope with the projected changes, both short and long-term adaptation strategies must be timely planned by the sector stakeholders and decision makers to adapt to a warmer and dryer future [20]. A good solution could be the use of kaolin products that significantly increase the polyphenolic profile of the leaves as reported in [21]. In this study, the effect of this product was determined to influence the quality of the olives and the oil. Several studies in the literature have investigated the antiparasitic effect of kaolin on different crops [22] and its anti-transpirant effect through the formation of a white film on the leaf surface that increases the reflection of incoming solar radiation as well as reduces heat stress and solar injury to the entire tree canopy, leaves, and fruits [23].

The Campanian cultivar “Racioppella” is a minor cultivar from the Benevento province, characterized by large and elliptical leaves and drupes with medium size and ellipsoidal shape. It is a highly appreciated cultivar, especially for its constant productivity, oil quality, and its high rusticity despite not offering a high yield (10–12%). The drupes come off the branch with difficulty and have a gradual veraison. It is a cultivar resistant to abiotic stress such as low temperatures and drought [5]. The sensory profile of olive oil is characterized by a medium fruity, very clean, delicate, and sweet taste, with a light bitterness and pungency. The Racioppella cultivar is characterized by a ratio oleic (61.36%) and linoleic (14.36%) equal to 4.39 and by the low endowment of phenolic compounds [24]. This study aimed to evaluate the effect of the application of kaolin in the production and quality parameters of the drupes and olive oil.

## 2. Results

### 2.1. Effects of Kaolin on Drupe Characteristics

The maximum, minimum, and average temperatures during the test period in “Castelvenere” (BN) city are shown in Figure 1. These data confirm the increase in temperatures recorded during the growing season; in fact, in mid-August there were temperatures of around 42 °C with an average of around 30 °C, confirming that the plants had been subjected to high temperatures during the experiment.

In Table 1, the Jaen index, production/plant, oil content, anthocyanins, carotenoids, and total chlorophyll in the drupes are shown. The olives were harvested at the end of October when the drupes had a Jaen index of 2.87 and 3.91, respectively, for the control thesis (C) and kaolin (K) (Figure 2). The treatment with kaolin showed a significant highest oil content compared with the control of about 3.4%, but it did not show effects on production per plant.

Kaolin treatment reported a greater veraison of the drupes with an increase in anthocyanins (531.06 mg/kg vs. 428.03 mg/kg) of about 24%. The concentrations of total chlorophyll were 12.52 mg/kg for K and 15.88 mg/kg for C, while the carotenoids contents were 6.78 mg/kg in C and 4.17 mg/kg in K. Therefore, for the carotenoids and total chlorophyll content, a lower concentration in drupes treated with kaolin was shown with a reduction of 38% for carotenoids and of 21% for total chlorophyll.

### 2.2. Polyphenolic Compounds Analysis by UHPLC-Q-Orbitrap HRMS and Antioxidant Activity of Olive Drupes

Table 2 shows the effect of treatment with kaolin-based anti-transpirant (K) on the qualitative and quantitative profile of polyphenolic compounds in olive drupes compared with the control (drupes from untreated plants).

The results obtained showed a significant increase with K application in the content of flavonoids (rutin and luteolin, +~70% and +~161%, respectively), phenolic acids (vanillic acid, +~122%), phenolic alcohols (hydroxytyrosol glucoside, hydroxytyrosol, and tyrosol, +~87%, +~110%, and +~66%, respectively) and secoiridoids (verbascoside, DHPEA-EDA, ligstroside, oleuropein, p HPEA-EDA, hydroxy-oleuropein-aglycon, 3,4-DHPEA-AC, DHPEA-EA, and p-HPEA-EA, +~54%, +~6%, +~108%, +~26%, +~61%, +~23%, +~234%, +~52%, and +~294%, respectively), compared with untreated fruits (C). The application with the anti-transpirant product reported an increase in total polyphenols of about 60% compared with the control. Free radical scavenging activity (DPPH), ferric reducing antioxidant activity (FRAP), and ABTS-scavenging activity are reported in Figure 3.

In accordance with the results obtained from mass spectrometry investigations, the antioxidant activity evaluated with all three measurement methods was higher in fruits harvested from plants treated with kaolin-based anti-transpirant with an average increase, compared with the control, of 41% and with the highest increase observed in the case of free radical-scavenging activity of ABTS and DPPH (+42.6% in both cases). ABTS, DPPH, and FRAP patterns were relatively similar to that observed in total polyphenol contents, being likely to contribute to the antioxidant activity of the extracts.

### 2.3. Effects of Kaolin on Oil Quality

In Table 3 the results of the quality indices of olive oil are shown.

No statistical difference for acidity was found (0.28% in C and 0.29% in K), while peroxide value was higher in K (6.86 meq O_2_/kg) than in C (5.34 meq O_2_/kg). K_270_ and K_232,_ K_232_ absorption coefficients, were higher in K than in C (0.21 vs. 0.20 for K_270_ and 1.64 vs. 1.46 for K_232_). However, all the values of the quality indices of the two oils analyzed were within the values that define an extra virgin olive oil.

In Table 4, the panel test was shown. Both the oil samples showed median of defects 0 and median of fruity > 0, so the oils must be considered as “Extra virgin olive oil” in accordance with EC Regulation 2019/1604. Furthermore, the control sample showed the highest value in terms of fruity, bitter, and spicy attributes compared with oil obtained after kaolin treatment.

### 2.4. Fatty Acids Composition of Olive Oil

The fatty acids composition of olive oil is shown in Table 5.

The treatment with kaolin increased the content of monounsaturated fatty acids (MUFAs) (71.13% vs. 67.34%), while it decreased the content of (polyunsaturated fatty acids) PUFAs (13.01 vs. 15.14%) and saturated fatty acids (SFAs) (15.87% vs. 17.53%). Among MUFAs, the oleic acid is the most present with a higher content in K than in C (70.22% vs. 66.23%) with an increase of 6%, while among SFAs, the treatment with kaolin reduced the concentrations of palmitic and stearic acids (12.56% vs. 13.90% for palmitic acid and 2.64% vs. 2.90% for stearic acid). Therefore, the kaolin treatment increased the concentration of MUFAs (+5.6%) and reduced SFAs (−9.46%). Among PUFAs, the linoleic acid is most present in C compared with K (14.06% vs. 11.82%); the kaolin reduced its concentration of 16%. Furthermore, the ratios of MUFAs/PUFAs and MUFAs/SFAs, and the oleic acid to linoleic acid ratio, were calculated. The ratios between MUFAs/PUFAs and MUFAs/SFAs were higher in K than in C consequently due to the lower value of PUFAs and the higher value of MUFAs in K than in C; finally, the oleic acid/linoleic acid ratios were higher in the K sample than in C (5.94 vs. 4.71).

### 2.5. Polyphenols Content of Olive Oil

The total polyphenols content (TPC) of the oil is shown in Table 6.

The TPC was higher in sample K than in C (489.80 mg GAE/kg vs. 442.31 mg GAE/kg) and the TPC increased to 11% after the anti-transpirant application. Regarding the individual polyphenols, higher contents of both tyrosol and *p*-coumaric acid were shown in K compared with C (9.85 vs. 8.17 mg/kg and 4.55 vs. 4.31 mg/kg for tyrosol and *p*-coumaric acid, respectively). The vanillic and ferulic acid content was higher in C than in K (0.42 mg/kg vs. 0.31 mg/kg and 0.81 mg/kg vs. 0.81 mg/kg, respectively).

## 3. Discussion

### 3.1. Effects of Kaolin on Drupe Bioactive Characteristics

The darkening into the purple-black of the pulp in the drupes happens in concomitance to the increase in the oil content [25] in agreement with our results that reported an increase in both parameters with kaolin application. Our results regarding the drupes oil content are in agreement with those reported by Saour and Makee [26], which showed an increase in oil content after the treatment with kaolin while disagreeing with those described by Arafat and Sayed [27] that reported a reduction in oil content after the treatment with kaolin. One of the attributes for judging maturity is skin color. In general, during fruit ripening, the chlorophylls, which are present in all unripe fruit, break down following the transformation of chloroplasts into chromoplasts [28]. As ripening progresses, photosynthetic activity decreases and the concentrations of both chlorophylls and carotenoids decrease progressively. At the end of the maturation process, the violet or purple color of the olive fruit is due to the formation of anthocyanins [25]. The maturation stage of collected olive fruits is a very important parameter, since it affects olive oil quality, stability, and composition [29]. In accordance with our results (Table 1), Wang et al. and Brillante et al. [30,31] showed a positive effect of kaolin treatment on the anthocyanins content of grapes and grapewines, respectively. Contrary to our results, Khavary et al. [32] showed that the kaolin treatment increased the contents of carotenoids and total chlorophyll in hazelnuts, while Brito et al. [33] found no significant difference between kaolin and control treatment in carotenoids and chlorophyll contents in the leaves of olive plants. Criado et al. [25] showed a concentration of total chlorophyll in a range between 3.1 mg/kg to 285 mg/kg in the Arbequina cultivar and between 1.6 mg/kg to 392 mg/kg in the Farga cultivar, depending on the ripening stage. Regarding the carotenoids content, Motilva and Romero [34] showed a concentration that ranged between 1.8 mg/kg d.w to 70 mg/kg d.w, depending on both cultivar and maturation time. Kaolin applications on olive trees did not alter olive and increased fruit yield in agreement with Roussos et al. [35].

Olives and virgin olive oil are valuable sources of natural phenolic antioxidants that have beneficial effects on human health [12,36]. In the bibliography, there are several studies about the effect of kaolin on the phenolic and antioxidant components in drupes. Our results are partially in agreement with those reported by Brito et al. [37] which emphasized that the effect of the treatment with kaolin is strictly dependent on climatic conditions. The authors’ results showed an increase in the concentration of compounds’ phenolic and antioxidant capacities in both fruits and oil only in the first year while a decrease in these parameters was observed in the second year in close association with the prevailing climatic conditions because they determine different responses in the plants. In our previous work, the significant enhancing effect of kaolin treatment on the nutraceutical potential of olive leaves in water deficit stress conditions was reported in terms of total phenolics and antioxidant activities [38]. On the other hand, Hagagg and his co-workers [39] reported that spraying kaolin and calcium carbonate at a concentration of 7% improved the vegetative growth parameters of the olive cultivar, and at the same time, Dinis et al. [40] showed that kaolin application may reduce the ROS levels and enhance the antioxidant defenses of plants including phenolics, flavonoids, anthocyanin, and all key metabolites in leaves and fruits of grapevine. The increase in antioxidant capacity due to the application of kaolin was also observed by Dinis et al. [40]; the authors investigated this effect by studying changes in the enzymatic and nonenzymatic antioxidant systems in leaves and berries of grapevine under summer stress. No significant effects of kaolin treatment were observed by Cosìc et al. [41] on sweet peppers, whereas kaolin treatments positively affected the antioxidant activity and contents of some individual phenolics on hazelnut, in trees with and without irrigation [42]. Finally, a 6% kaolin spray was effective in improving the contents of phenolics and flavonoids, and the antioxidant enzyme activity of mango fruits [43]. It should be noted that the kaolin application is strongly correlated to climatic conditions since kaolin is a product that washes away with the rain, and in fact, in the bibliography, various studies reported repeating the treatment after heavy rains [37].

### 3.2. Effects of Kaolin on Olive Oil Quality

Extra virgin olive oil (EVOO) is responsible for a large part of many health benefits associated with the Mediterranean diet as it is a fundamental ingredient of this diet. The peculiarities of this golden, highly valued product are in part due to the requirements that must be met to achieve this title, namely, it has to be obtained using exclusively mechanical procedures, its free acidity cannot be greater than 0.8%, it must not show sensory defects, and it has to possess a fruity taste [44]. In this study, the oil from untreated drupes showed the highest value in terms of fruity taste, bitterness, and spiciness, highlighting that treatment with kaolin reduced the oil-positive sensorial intensity. Brito et al. [45] showed, conversely, that the oil quality indices were not influenced by the treatment, and the K_270_ coefficient was reduced by kaolin application; moreover, Rotondi et al. [46] showed no significant influence of kaolin treatment on the quality parameters of oil. On the contrary, in other studies it was shown that the sensory attributes of oil were unaffected by kaolin [47]. Furthermore, Rotondi et al. [48] showed that the kaolin treatment reduced the intensity of olfactory fruitiness, but it increased its bitterness. The beneficial properties of olive oil are mainly attributed to its composition, a high percentage of monounsaturated acids (oleic acid: C18:1, n 9), and significant amounts of minor components with strong antioxidant activity. The most present fatty acids in both oils analyzed were oleic, palmitic, and linoleic acids, as reported in other studies [11,49]. In general, the fatty acids composition is in accordance with the range proposed by IOC regulation; however, a small exception was found but that could be the result of genetic and agronomic factors [49,50]. In general, in our study the total MUFAs and PUFAs were similar to the results of Sakar et al. [51] that showed a range from 69.53% to 75.45% for MUFAs and from 11.06% to 15.16% for PUFAs, depending on the oil extraction system used. Our results are in accordance with Khaleghi et al. [52] that also found an increase in oleic acid and reductions in palmitic, stearic, and linoleic acids in olive oil treated with kaolin. On the other hand, our results conflict with those described by Rotondi et al. [46] that showed an increase in linoleic acid content after the kaolin treatment, while no significant differences in oleic, palmitic, and stearic acids were found. Linoleic acid decreases shown in our study were confirmed by Khaleghi et al. [52] and Arafat and EL-Sayed [27] who showed a reduction in this fatty acid, while oleic acid was most present in K compared with C (1.19% vs. 1.08%). Our result was in contrast with Khaleghi et al. [52] and Arafat and EL-Sayed [27] who reported an increase in oleic acid after kaolin treatment. Finally, the ratio of oleic acid/linoleic acid was in the range of Piscopo et al. [53] who reported a value ranging from 4.78 to 12.18 depending on the analyzed cultivar.

Brito et al. [45] showed a positive effect of kaolin treatment depending on the climatic condition of plants grown. Furthermore, Khaleghi et al. [52] showed the total polyphenols content ranged between 424 mg GAE/kg to 554 mg GAE/kg depending on the percentage of kaolin particle film used, and Rotondi et al. [46] showed an increase in TPC in olive oil after the foliar treatment with kaolin, in agreement with our results. Furthermore, Graziani et al. [38] showed the highest TPC in the leaves of olive trees that were treated with kaolin. Regarding the individual polyphenols, the tyrosol content was in accordance with the results of Torić et al. [54] that found a range between 4.57 mg/kg to 9.26 mg/kg depending on the cultivar of analyzed olive oil and with the results of Murkovic et al. [55] that showed a content that ranged between 1.4 mg/kg to 29 mg/kg. The vanillic acid content was reduced by anti-transpirant treatment (Table 6). However, both oils showed a lower content compared with what Torić et al. [54] and Murkovic et al. [55] reported since a concentration of vanillic acid in the range of 0.50 mg/kg to 0.95 mg/kg, and between 0.67 mg/kg to 4.0 mg/kg, respectively, was shown.

Finally, the *p*-coumaric acid was higher than the results of Ogras [56], which showed values between 0.11 mg/kg to 0.20 mg/kg; compared with our results, he also showed that ferulic acid was highest with a concentration that ranged between 1.04 mg/kg to 1.20 mg/kg. Orgas [56] reported that the concentration of phenolic compounds in olive oil is dependent on the cultivar, harvest year, and geographic region.

## 4. Materials and Methods

### 4.1. Experimental Design and Plant Materials

The test was conducted in an olive orchard in Castelvenere near Benevento (Southern Italy) (41°23′ N, 14°55′ E; 140 m a.s.l.) during the growing season from May to October 2021. The experiment was carried out on olive trees of about 20 years old belonging to the “Racioppella” cultivar. The plants were trained to have an open-vase shape and spaced 6.00 m between the rows, and 5.00 m in the rows. The trial was organized with a randomized block experimental design (3 blocks of 4 plants each) for a total of 12 plants per treatment. The experimental design was based on kaolin treatments compared with the control:(1)Control (C) plants were only treated with water.(2)Kaolin (K): the product used was Manisol (kaolin + copper 5%) from Manica S.p.a (Rovereto, Italy), which was used by foliar application with 3.5 kg/hL.

Kaolin was applied 3 times during the growing season at 30-day intervals, from July to September, at the 71, 79, and 81 BBCH scale phenological stages [57]. The temperatures recorded during the growing season were downloaded from the Castelvenere meteorological station where the study was conducted.

### 4.2. Harvest, Production/Plants, Maturation Index, and Oil Extraction

The olives were harvested at the end of October by a trunk shaker applied to 12 plants per harvest, when about 50% of the fruits reached the veraison. The evaluation of maturation was conducted through an assessment of the pigmentation of the olives (Jaen’s index 0–7). Fruits were weighed to determine the yield per plant by a digital dynamometer (Kern&Sohn, model 50K50—Baden-Württemberg, Germany) and they were transported to a crusher to be processed with a 3-phase continuous malaxing machine (Piralisi F.lli Spa.—Ancona, Italy).

### 4.3. Fat Extraction by Drupes

The fat content was extracted following the method of Gonçalves et al. [58], with modifications. Briefly, 4 g of olive was added to 100 mL solution of chloroform/methanol (2:1; *v*/*v*) (Carlo Erba reagents, Milan, Italy) and 100 mg L^−1^ of butylated hydroxytoluene (BHT) (Sigma-Aldrich St. Louis, MO, USA). Then, a homogenization was performed using an ultra-turrax (Janke and Kunkel—Baden-Württemberg, Germany, type TP 18/10) for 6 min on ice. The extract was filtered and added to a separating funnel and the procedure was repeated twice. The obtained volume was adjusted to 150 mL with chloroform/methanol (2:1; *v*/*v*) and 37.5 mL of sodium chloride (0.73%) was added. After mixing, it was left to rest for 20 min. Then, the lipidic extract was recovered and filtered with sodium sulfate anhydrous (Na_2_SO_4_) (Sigma-Aldrich St. Louis, MO, USA). The lower phase was collected from previously weighted glass flasks and the solvent was evaporated using a rotary evaporator.

### 4.4. Carotenoids and Chlorophylls Determination of the Drupes

The carotenoids and chlorophylls determination on drupes was performed following the spectrophotometric method reported by Aiello et al. [59], with modifications. The sample (100 mg) was dissolved in 5 mL of ethyl ether and then placed in an ultrasonic bath for 1 min and vortexed for 30 s. The absorbance was measured by using a Shimadzu UV-1601 spectrophotometer (Shimadzu, Kyoto, Japan) at a wavelength of 470 nm for carotenoids and at 660 nm and 643 nm for chlorophyll *a* and *b*, respectively. The results are expressed as mg per kg.

### 4.5. Anthocyanins Determination of the Drupes

The anthocyanins content on drupes was determined following the method of Raj and Ahmad [60], with modifications. Briefly, 2 g of fruit epicarp was macerated in 20 mL of 5% acidified methanol using a mortar and pestle. The extraction was repeated three times. The obtained extract was centrifuged at 6500 rpm for 10 min. The supernatant was kept in a dark environment overnight. Finally, the absorbance was measured at 520 nm using a Shimadzu UV-1601 spectrophotometer (Shimadzu, Kyoto, Japan).

The absorbance measurements were performed using a Shimadzu UV-1601 spectrophotometer (Shimadzu, Kyoto, Japan) at wavelength of 520 nm. The total anthocyanin was expressed as mg of cyanidin-3-glucoside equivalent per kg.

### 4.6. Ultrasound-Assisted Extraction of Polyphenolic Compounds from Drupes

Phenolic standards were purchased from Sigma Aldrich St. Louis, MO, USA, while hydroxytyrosol was obtained from Indofine (Hillsborough, NJ, USA), secologanoside from ChemFaces Biochemical Co., Ltd. (Wuhan, China), and oleuropein from Extrasynthese (Genay, France). Stock solutions of olive oil phenolics were prepared at 1 mg mL^−1^ in methanol and stored at −20 °C for 1 month. For each standard calibration, curves were built in the range 0.02–5 mg mL^−1^. Methanol, hexane, and formic acid (LC-MS grade) were obtained from Carlo Erba reagents (Milan, Italy), whereas acetic acid (98–100%) was supplied from Fluka (Milan, Italy).

Lyophilized drupes were ground in a mill IKA A11 (IKAWerke, Staufen, Germany) and then extracted using the procedure reported in the literature [61] with minor modifications. In particular, 0.2 g of dried sample were extracted with 3 mL of a mixture of methanol/water (80:20 *v*/*v*, 0.1% formic acid) by sonication at room temperature for 15 min. After the mixture was centrifuged to 4000 rpm at 4 °C for 10 min, the supernatants were collected and filtered through 0.45 mm nylon syringe membranes. The final extract was washed with 4 mL of n-hexane and dried under nitrogen flow. The dried phenolic extracts were solubilized in 1 mL of methanol before high-resolution mass spectrometry analysis and antioxidant activity tests.

### 4.7. UHPLC-HRMS Analysis of Polyphenolic Compounds and Antioxidant Activity Evaluation of the Drupes

Phenolic compounds were quantified and separated using an UHPLC system (Thermo Fisher Scientific, Waltham, MA, USA) equipped with Kinetex 1.7 µm Biphenyl (100 mm × 2.1 mm) column (Phenomenex, Torrance, CA, USA) according to the conditions described by Cirillo et al. [22]. Mass spectrometry analysis was conducted on a Q Exactive Orbitrap LC-MS/MS (Thermo Fisher Scientific, Waltham, MA, USA). The acquisition of polyphenolic compounds was performed according to Dini et al. [62] where the analytical conditions are fully detailed. A mass tolerance of 5 ppm was employed. The instrument calibration was checked daily using a reference standard mixture obtained from Thermo Fisher Scientific. The free radical scavenging activity of the polyphenolic extracts was analyzed using 2,2-diphenyl-1-picryl-hydrazyl (DPPH), ferric reducing antioxidant activity was measured using the FRAP assay, and finally the ABTS-scavenging activity was evaluated employing the ABTS (2,2′-azino-bis-3-ethylbenzothiazoline-6-sulfonic acid) method as detailed in Graziani et al. [63]. In particular, the radical scavenging activity of the drupes and oils extracts was determined by adding l mL of DPPH radical working solution and 200 µL of suitably diluted extract. The decrease in absorbance of the resulting solution was monitored at 517 nm after 10 min of incubation and the results are expressed in TEAC (mmol Trolox equivalents per kg dry weight of sample). All determinations were performed in triplicate. For FRAP (ferric-reducing antioxidant capacity) measurements, the extracts (150 µL) were allowed to react with 2.850 mL of FRAP reagent. The absorbance was monitored after 4 min at 593 nm. The results were expressed as TEAC (mmol Trolox equivalents per kg dry weight of sample). All the determinations were performed in triplicate. The ABTS assay was conducted by adding 100 mL of suitably diluted sample and 1 mL of the ABTS working solution. After 3 min, the absorbance was measured at 734 nm. The results are expressed as TEAC (mmol Trolox equivalents per kg dry weight of sample). All determinations were performed in triplicate.

### 4.8. Polyphenols Determination by HPLC and Total Polyphenols Content (TPC) of the Oil (FOLIN Assay)

The individual polyphenol concentration was evaluated by HPLC analysis. The sample (150 mg) was mixed with 3 mL of methanol. The mixture was shaken for 30 s and it was sonicated for 20 min and filtered with a 0.22 µm PES filter before injection into the HPLC system. The HPLC method was the same as that of Romano et al. [29]. To quantify the concentration of compounds, standard calibration curves (tyrosol, *p*-coumaric acid, ferulic acid, vanillic acid, and oleuropein) were constructed. The range of linearity was 1–200 ppm for tyrosol, *p*-coumaric acid, vanillic acid, and oleuropein, and 1–100 ppm for ferulic acid. The square of the correlation coefficient (R^2^) was 0.9985 for naringin, 0.996 for tyrosol, 0.997 for *p*-coumaric acid, 1 for ferulic acid, 0.994 for vanillic acid, and 0.993 for oleuropein. The limit of detection (LOD) and limit of quantification (LOQ) were 0.5 and 1 ppm, respectively. The results are expressed as mg per kg of oil.

The extraction and quantification of total polyphenols content (TPC) by the Folin–Ciocalteau method was performed following the procedure proposed by Genovese et al. [64], with modifications. The oil (300 mg) was dissolved in 300 µL of hexane, and the mixture was mixed for 30 s. Subsequently, 1.5 mL of methanol:water (60/40 *v*/*v*) was added to the mixture and it was vortexed for 1 min and centrifuged at 4000 rpm for 10 min. This procedure was repeated twice.

To the obtained extract (100 µL), 400 µL of water, 800 μL of 7.5% sodium carbonate (Na_2_CO_3_), and 100 µL of Folin–Ciocalteau (2 N) were added. The samples were left for 30 min at room temperature in the dark. The absorbance was measured at 750 nm wavelength by using a UV-1601 spectrophotometer (Shimadzu, Kyoto, Japan). The TPC was calculated by using a calibration curve of gallic acid and the results are expressed as mg of gallic acid per kg of oil. The analyses were performed in triplicate for each extraction.

### 4.9. Quality Indices of Olive Oil

The oil: acidity (% oleic acid per 100 g oil), peroxide value (meq O_2_ kg^−1^ oil), and spectrophotometric indices (K232, K270, and ΔK) are quality indices of olive oil and they were evaluated according to the EC Reg. 2568/1991 and later, with amendments, and the International Olive Council (IOC) standard methods. The sensory analysis was carried out by eight assessors who were fully trained in the evaluation of extra-virgin olive oil (EVOO) according to the official methods of the IOC (1996) and EC Reg. 1604/2019. The panel test was performed at the Department of Agriculture, University of Naples Federico II (Naples, Italy), using the evaluation form regulated by EC Reg. 640/2008.

### 4.10. Fatty Acid Profile of the oil

The oil fatty acid profile was evaluated by analyzing the fatty acid methyl esters (FAMEs) obtained after trans-esterification, following the procedure proposed by Di Vaio et al. [11]. The obtained extract (1 µL) containing the FAMEs was injected into an Agilent Technologies 6890N gas chromatograph equipped with a capillary column. The characteristics of the column and the GC method were the same as those of Di Vaio et al. [11]. The identification of the compounds was performed by comparison with a mixture of standards: FAME C4–C24 (Sigma-Aldrich fatty acid methyl esters 24 components). The results are expressed as % *w*/*w*.

### 4.11. Statistical Analysis

Analysis of variance (ANOVA) was applied to analyze the means. Student’s *t*-test was applied to compare the means between two groups (*p* < 0.05). The statistical package XLSTAT Version 2013 (New York, NY, USA) was implemented for all the analyses.

## 5. Conclusions

This study highlighted the efficiency of an anti-transpirant product in improving the quality of both the drupes and the oil of olive. In fact, kaolin application increased the anthocyanin content of the drupes (+24%), the total polyphenols content (+60%), and reported significant values in increasing the antioxidant activity of drupes (+41%). Regarding the olive oil, kaolin application reported an increase in monounsaturated fatty acids, oleic and linoleic acids, and confirmed an increase in total polyphenols. Based on our study, it is possible to state that kaolin is a valid solution for sustainable agriculture; it is a valid product both for its known activity against pests and for improving the qualitative parameters of olive plants in actual environmental conditions and for improving the qualitative parameters of both the drupes and the oil.

## Figures and Tables

**Figure 1 plants-12-01259-f001:**
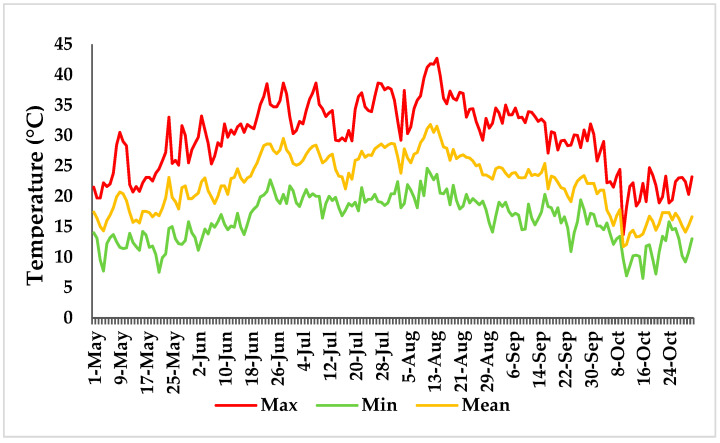
Temperature variation: maximum, minimum, and mean temperature (°C) recorded during the growing season in the open field (from May to October in “Castelvenere”—BN).

**Figure 2 plants-12-01259-f002:**
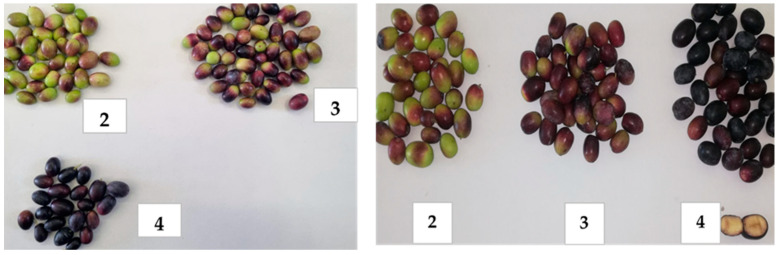
Determination of epicarp color of untreated (**left**) and kaolin-treated (**right**) drupes at harvest with the Jean index (2: yellow skin with red spots, 3: light red or light purple skin,4: black skin and green flesh).

**Figure 3 plants-12-01259-f003:**
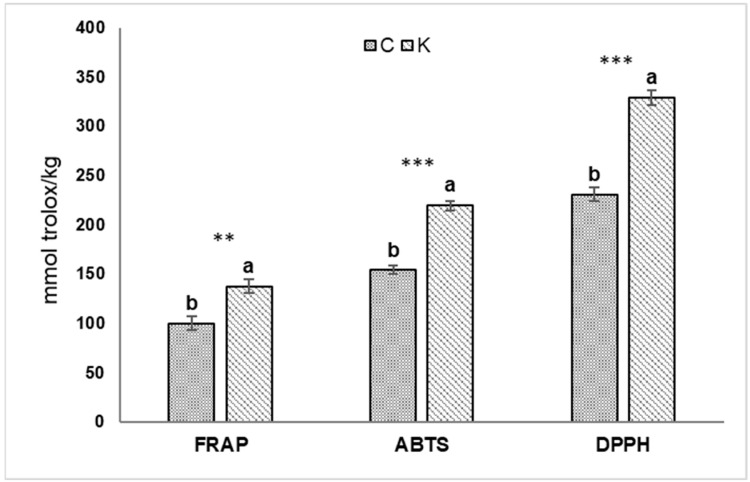
Antioxidant activity (FRAP, ABTS, and DPPH) in olive drupe treated with kaolin product (K) compared with control (C). Values are mean ± standard error and different letters indicate significant differences based on Student’s *t*-test (*p* = 0.05). Asterisks indicate a significant effect of kaolin treatment according to ANOVA (** = *p* < 0.01; *** = *p* < 0.001).

**Table 1 plants-12-01259-t001:** Production/plant, Jean index, oil content (%), anthocyanins (mg/kg), carotenoids (mg/kg), and total chlorophyll at the time of drupe harvest treated with kaolin product (K) compared with the control (C).

	C	K	Significance
Jaen index	2.87	3.91	
Production/plant (kg)	25.00 ± 0.95 a	25.30 ± 0.44 a	ns
Oil content (% f.w.)	14.30 ± 0.14 b	14.78 ± 0.02 a	*
Anthocyanins (mg/kg)	428.03 ± 4.57 b	531.06 ± 2.13 a	***
Carotenoids (mg/kg)	6.78 ± 0.08 a	4.17 ± 0.10 b	***
Total chlorophyll (mg/kg)	15.88 ± 0.18 a	12.52 ± 0.38 b	**

Values are mean ± standard error. Different letters indicate significant differences according to Student’s *t*-test (*p* = 0.05). Asterisks indicate significant effect of kaolin treatment according to ANOVA (ns = not significant; * = *p* < 0.05; ** = *p* < 0.01; *** = *p* < 0.001).

**Table 2 plants-12-01259-t002:** Phenolic profiles and total polyphenols in olive drupes treated with kaolin product (K) compared with control (C). Concentrations are expressed as mg/g d.w.

	C	K	Significance
Hydroxytyrosol glucoside	4.85 ± 0.47 b	9.05 ± 0.51 a	***
Hydroxytyrosol (3,4-DHPEA)	73.18 ± 7.10 b	153.07 ± 22.05 a	*
Tyrosol (4-HPEA)	8.26 ± 1.12 b	13.75 ± 1.82 a	*
Vanillic acid	7.52 ± 1.03 b	16.66 ± 3.40 a	*
Rutin	56.35 ± 16.64 b	95.61 ± 22.83 a	**
Elenolic acid	20.25 ± 2.17 a	18.61 ± 1.29 a	ns
Verbascoside	7078.61 ± 618.05 b	10,872.06 ± 482.56 a	**
DHPEA-EDA	284.12 ± 133.34 a	300.93 ± 18.25 a	ns
Ligstroside	47.00 ± 23.97 b	98.03 ± 25.04 a	**
Oleuropeina	442.56 ± 97.58 a	559.36 ± 103.44 a	ns
p HPEA-EDA	10.51 ± 1.35 b	16.89 ± 0.59 a	***
Hydroxy–Oleuropein–aglycon	5.52 ± 1.66 a	6.79 ± 0.89 a	ns
Luteolin	14.92 ± 0.90 b	38.91 ± 8.05 a	*
3,4-DHPEA-AC	12.27 ± 1.84 b	41.02 ± 3.16 a	***
DHPEA-EA	471.32 ± 288.33 b	718.48 ± 213.55 a	**
p-HPEA-EA	7.60 ± 2.82 b	29.99 ± 4.55 a	**
Total polyphenols	8102.26 ± 679.24 b	12,989.23 ± 241.28 a	***

Values are mean ± standard error. Different letters indicate significant differences according to Student’s *t*-test (*p* = 0.05). Asterisks indicate significant effect of kaolin treatment according to ANOVA (ns = not significant; * = *p* < 0.05; ** = *p* < 0.01; *** = *p* < 0.001).

**Table 3 plants-12-01259-t003:** Quality indices of analyzed oils with kaolin treatment (K) compared with control (C).

Oil Quality Index	C	K	Significance
Acidity (% oleic acid)	0.28 ± 0.01 a	0.29 ± 0.02 a	ns
Peroxide value (meq O_2_/kg)	5.34 ± 0.03 b	6.86 ± 0.01 a	***
K_270_	0.20 ± 0.00 b	0.21 ± 0.00 a	*
K_232_	1.46 ± 0.00 b	1.64 ± 0.01 a	***
ΔK	0.00 ± 0.00 a	0.00 ± 0.00 a	ns

Values are mean ± standard error. Different letters indicate significant differences according to Student’s *t*-test (*p* = 0.05). Asterisks indicate significant effect of kaolin treatment according to ANOVA (ns = not significant; * = *p* < 0.05; *** = *p* < 0.001).

**Table 4 plants-12-01259-t004:** Results of oils panel test with kaolin treatment (K) compared with control (C).

Panel Test	C	K
Fruity	7.2	5.4
Bitter	3.6	3
Spicy	4.6	2.4
Heating/sludge	0	0
Winey/acid/acidic/sour	0	0
Rancid	0	0
Mold/moisture/ground	0	0
Frozen olive	0	0

**Table 5 plants-12-01259-t005:** Fatty acid composition of oil with kaolin treatment (K) compared with control (C).

% Fatty Acids	C	K	Significance
Palmitic (C16:0)	13.90 ± 0.06 a	12.56 ± 0.02 b	***
Palmitoleic (C16:1)	1.10 ± 0.01 a	0.91 ± 0.04 b	*
Heptadecanoic (C17:0)	0.20 ± 0.00 a	0.19 ± 0.00 a	ns
Stearic (C18:0)	2.90 ± 0.02 a	2.64 ± 0.01 b	**
Oleic (C18:1)	66.23 ± 0.00 b	70.22 ± 0.05 a	***
Linoleic (C18:2)	14.06 ± 0.04 a	11.82 ± 0.00 b	***
Arachidic (C20:0)	0.39 ± 0.00 a	0.36 ± 0.00 b	**
Linolenic (C18:3)	1.08 ± 0.00 b	1.19 ± 0.01 a	**
Behenic (C22:0)	0.14 ± 0.00 a	0.13 ± 0.00 a	ns
MUFA	67.34 ± 0.01 b	71.13 ± 0.02 a	**
PUFA	15.14 ± 0.04 a	13.01 ± 0.01 b	***
SFA	17.53 ± 0.04 a	15.87 ± 0.00 b	***
MUFA/PUFA	4.38 ± 0.01 b	5.40 ± 0.01 a	***
MUFA/SFA	3.78 ± 0.01 b	4.43 ± 0.00 a	***
Oleic/linoleic	4.71 ± 0.01 b	5.94 ± 0.01 a	***

Values are mean ± standard error. Different letters indicate significant differences according to Student’s *t*-test (*p* = 0.05). Asterisks indicate a significant effect of kaolin treatment according to ANOVA (ns = not significant; * = *p* < 0.05; ** = *p* < 0.01, *** = *p* < 0.001).

**Table 6 plants-12-01259-t006:** Polyphenols content of oil with kaolin treatment (K) compared with control (C).

Polyphenols (mg/kg)	C	K	Significance
Tyrosol	8.17 ± 0.11 b	9.85 ± 0.10 a	***
*p*-coumaric acid	4.31 ± 0.01 b	4.55 ± 0.00 a	***
Ferulic acid	0.81 ± 0.01 a	0.76 ± 0.01 b	*
Vanillic acid	0.42 ± 0.01 a	0.31 ± 0.01 b	**
Oleuropein	<LOD	<LOD	
TPC (mg GAE/kg)	442.31 ± 10.77 b	489.80 ± 0.02 a	**

Values are mean ± standard error. Different letters indicate significant differences according to Student’s *t*-test (*p* = 0.05). Asterisks indicate significant effect of kaolin treatment according to ANOVA (* = *p* < 0.05; ** = *p* < 0.01; *** = *p* < 0.001).

## Data Availability

Not applicable.

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
