# Peer review of "Minor Variety of Campania Olive Germplasm (“Racioppella”): Effects of Kaolin on Production and Bioactive Components of Drupes and Oil"

_plants, 2023, doi:10.3390/plants12061259_

Round 1
Reviewer 1 Report
Line 26 – instead of „say“ write „conclude“
Line 27 – write „in olive drupes an oil“
In the whole text – write properly the temperature with space, for ex. „30 °C“, not „30°C“; and the percentages as well: „38%“, not „38 %“
Line 143 – anti-transpirant
Line 224 – delete „didn't“
Line 227 and so on – why is the word „cultivar“ in italic?
Line 233 „and virgin olive oil“
Line 266 – delete „the“
Line 277 – correct the reference by putting an ordinal number
Line 306 – delete „that reported“
Line 307 and in the whole text – Kg – not with uppercase
Line 316 – „41°23’ N.“ delete the fullstop
Line 316 - correct in „m.a.s.l.“
Line 316 – delete 2021
Line 324 – correct in „3.5 kg /hL“
Line 337-338 – „The fat content was extracted following the method of Gonçalves et al.[58], with modifications.“ – what kind of modifications? Be specific.
Line 346 – delete „using“
Line 365 - instead of „shall be“ write „was“
Line 404 – instead of „in“ write „at“
Line 453 – correct the word “ tooil“
Author Response
Reviewer 1
Dear Reviewer, thank you for your corrections. All changes have been reported in the text.
- Line 26 – instead of „say“ write „conclude“
- It has been replaced in the text.
- Line 27 – write „in olive drupes an oil“
- It has been modified.
- In the whole text – write properly the temperature with space, for ex. „30 °C“, not „30°C“; and the percentages as well: „38%“, not „38 %“
- It has been modified.
- Line 143 – anti-transpirant
- Done
- Line 224 – delete „didn't“
- Done
- Line 227 and so on – why is the word „cultivar“ in italic?
- It has been changed.
- Line 233 „and virgin olive oil“.
- It has been changed.
- Line 266 – delete „the“
- Done
- Line 277 – correct the reference by putting an ordinal number
- This reference has been removed.
- Line 306 – delete „that reported“
- It has been changed in “have reported.”
- Line 307 and in the whole text – Kg – not with uppercase
- Done
- Line 316 – „41°23’ N.“ delete the fullstop
- Done
- Line 316 - correct in „m.a.s.l.“
- Done
- Line 316 – delete 2021
- Done
- Line 324 – correct in „3.5 kg /hL“
- It has been changed.
- Line 337-338 – „The fat content was extracted following the method of Gonçalves et al.[58], with modifications.“ – what kind of modifications? Be specific.
- Done
- Line 346 – delete „using“
- Done
- Line 365 - instead of „shall be“ write „was“
- Done
- Line 404 – instead of „in“ write „at“
- Done
- Line 453 – correct the word “ tooil“
- It has been changed.

Reviewer 2 Report
This study looks for the oil quality caracteristics of a Local Olive tree conducted with and without kaolin treatment.
THe document is globally well written, some english expressions should be reviewed. A few more data should be provided in the material and method section.
More detailed comments can be found in the document attached

Author Response
Dear Reviewer, thank you for your corrections. The authors have included your changes in the text.
- Line 23: later in this abstract you say +11%%, please check and correct.
- Dear reviewer, the increase in polyphenols was 60% for drupes, while the increase was 11% for olive oil.
- Line 26: “we”
- It has been changed.
- Line 35: the number of cultivated varieties
- It has been changed.
- Line 40: already said in line 34.
- This sentence has been deleted.
- Line 46: do you main, the cultivation or presence of olive or Olea europaea crop?
- This sentence has been modified in “The use of Olea europaea, both as table and oil cultivars, is well-documented in these regions, with many archaeological and written relics dating back to ancient times, attested also by the presence of monumental trees”.
- Line 52 and 55: please explain the acronym.
- The meaning of the acronym is Protected Designation of Origin and Protected Geographical Indication. It has been added in the text.
- Line 58: varieties
- Done
- Line 78 – 92: It will be more convenient to stat this paragraph talking about the benefits kaolin may have on crops and focus in the analysis on the anti transpirant potential making the link with the previous paragraph.The finish with the combined objective of characterize the cultivar and the impact of kaolin.
- Dear Reviewer, Thank you for your advice. the authors have changed this section.
- Line 103: precise again the location of the recorded data.
- Done
- Line 123: were.
- Done
- Line 172: maybe add characteristics or oil attributes at the end of the sentence.
- Done
- Line 212: El
- The authors checked and edited the quote.
- Line 220: accordance
- Done
- Line 230: on
- Done
- Line 231: please check.
- The sentence has been changed to: “Kaolin applications on olive trees did not alter olive and the increased fruit yield in agreement with Roussos et 231 al. [35]” (243-244).
- Line 237: please consider the revision of this sentence, it's not clear enough:
- This sentence has been modified in: “Our results are partially in agreement with those reported by Brito et al [37] who emphasized that the effect of the treatment with kaolin is strictly dependent on climatic condition, determining an increase in the concentration of compounds phenolic and antioxidant capacity in both fruits and oil only in the first year while a decrease in these parameters is observed in the second year in a closely association with the prevailing climatic conditions because they determine different responses by the plants.”
- Line 248- 249: How you can discriminate the effect of kaolin and copper? Please justify in the material and method section whu you should add the copper.
- Dear reviewer, the product we used was a mixture with copper, but we checked the bibliography and there are no studies that demonstrate the efficiency of copper in increasing the antioxidant activity.
You could also discuss the impact of rain on the "quality" of the kaolin effect, and why you can find differences with other studies. And the differences on the thermical stresses plants could have suffered in the several studies.
- The authors added some information to line 272: "It is to underline that the kaolin application is strongly correlated to climatic conditions since kaolin is a product that washes away with the rain, in fact, various studies are reported in the bibliography in which the treatment is repeated after heavy rains [37]"
- Line 326 : at the 71, 79 and 81 BBCH scale phenological stages.
- It has been changed.
- Line 333: the model is missing.
- It has been added.
- Line 337: you need to explain at leat the modifictaions you have made, and explain briefly the method used.
- Done
- Line 341: again, any modification you have done should be indicated and also is convienent to provide a short description of the method used
- Done
- Line 343: could you precise which wavelength for carotenoids and which one for chlorophylls? because cited like this we image more a range of wavelengths
- Done
Line 346: same thing, add which modifications
- Done
- Line 350-357: this part should be placed at the beginning of this 4. Section - Line 359: you could group under the same section 4.7 & 4.8 and maybe 4.9 Polyphenol analysis and make 3 subsections to make it more coherent - Line 415: could join the 4.7 et other section, all under the Polyphenols Line 419 : same join the 4.7 section - Line 424: same join the 4.7 section
- Dear reviewer, thanks for the suggestions. Paragraphs were grouped as required.
- Line 382: Explain the acronym SIGNIFICATO ABTS E FRAP.
- The meaning of the acronym has been added in the text: ABTS (2,2′-azino-bis-3-ethylbenzothiazoline-6-sulfonic acid), FRAP ( Ferric-reducing antioxidant capacity) measurements.
- Line 424: indicate the acronym on the title.
- Done

Round 2
Reviewer 2 Report
The corrections request in the previous review have been done.
Now, the material and method section is complete and well organised
Author Response
Now the authors will upload the manuscript with the changes requested by the editor.
